# Effect of Jute Fiber Modification on Mechanical Properties of Jute Fiber Composite

**DOI:** 10.3390/ma12081226

**Published:** 2019-04-15

**Authors:** Hua Wang, Hafeezullah Memon, Elwathig A. M. Hassan, Md. Sohag Miah, Md. Arshad Ali

**Affiliations:** 1College of Textiles, Donghua University, 2999 North Renmin Road, Shanghai 201620, China; maadhu1987@yahoo.com; 2Donghua University Center for Civil Aviation Composites, Donghua University, 2999 North Renmin Road, Shanghai 201620, China; hm@mail.dhu.edu.cn (H.M.); elwathig2011@uofg.edu.sd (E.A.M.H.); 3Key Laboratory of Textile Science & Technology, Ministry of Education, College of Textiles, Donghua University, 2999 North Renmin Road, Shanghai 201620, China; sohagdonghua@gmail.com; 4State Key Laboratory for Modification of Chemical Fibers and Polymer Materials, Donghua University, Shanghai 200051, China; 5Industries Engineering and Technology, University of Gezira, Wad Madani 21111, Sudan

**Keywords:** jute fiber composite, chemical treatment, surface openness, physical properties, tensile properties

## Abstract

Recently, the demand for reinforced plastics from natural, sustainable, biodegradable, and environmentally friendly fibers has been rising worldwide. However, the main shortcoming of natural fibers reinforced plastics is the poor compatibility between reinforcing fibers and the matrix. Hence, it is necessary to form a strong attachment of the fibers to the matrix to obtain the optimum performance. In this work, chemical treatments (acid pretreatment, alkali pretreatment, and scouring) were employed on jute fibers to modify them. The mechanical properties, surface morphology, and Fourier transform infrared spectra of treated and untreated jute fibers were analyzed to understand the influence of chemical modifications on the fiber. Then, jute fiber/epoxy composites with a unidirectional jute fiber organization were prepared. Basic properties of the composites such as the void fraction, tensile strength, initial modulus, and elongation at break were studied. The better interfacial adhesion of treated fibers was shown by scanning electron microscope (SEM) images of fractured coupons. Hence, the chemical treatment of jute fiber has a significant impact on the formation of voids in the composites as well as the mechanical properties of jute fiber composites.

## 1. Introduction

During the last few decades, composite materials have gained much attention from researchers in the fields of materials science and engineering materials [1,2]. Broadly, composite materials can be categorized into three categories in terms of the matrix used: polymer matrix composites, metal matrix composites, and ceramic matrix composites. Among them, polymer matrix composites have various advantages over the other two, including a lower volume-to-weight ratio, a higher specific strength-to-weight ratio, the ability to be formed into different shapes and sizes, resistance to corrosion, as well as a simple manufacturing process, recyclability, and lower cost [1,3,4]. Therefore, fiber-reinforced plastics have successfully replaced their heavy metal and ceramic as well as expensive engineering plastic counterparts. In general, fiber-reinforced plastics are either made up of thermosets or thermoplastic resin as a matrix, with either synthetic or natural fibers as a reinforcing material [5,6,7]. Due to the large amounts of debris they generate and the scarcity of fossil fuels, more and more attention has been paid to the use of sustainable, biodegradable, and green composites. Natural fiber-reinforced composites fulfill these requirements, however, there is still much to be explored to gain the full benefit of these composites.

Jute is the second most natural and biodegradable fiber [8]. Jute fiber is an excellent alternative when strength, thermal conductivity, and cost are major concerns [8]. In addition, jute fibers are eco-friendly. Nowadays, jute fiber-reinforced polymer composites have become an important area of research [9,10,11,12,13,14]. Typically, jute fiber is used for basic and low-end textile products. If the properties of jute could be modified in favor of high value and technical textiles, not only the cost but also the environment would benefit a great deal. Jute is composed of cellulose (45–71.5%), hemicelluloses (13.6–21%), and lignin (12–26%) [13,14]. Lignin, due to the many aromatic rings inside of it, is responsible for mechanical support [13]. Any material besides cellulose that hampers the smoothness, pliability, and fineness of jute is denoted as gum [14].

Incompatibility between natural fibers and the matrix affects the interaction adhesion of the fiber with the matrix, which results in a poor fiber–matrix interface. The weak interface reduces the fiber’s reinforcing efficiency due to a lack of stress transfer from the matrix to the fibers. By opening up the cellulose content and removing unnecessary materials, chemical treatment makes fibers smooth (by removing the gum), easy to adhere, durable, and flexible and has a lasting effect on the mechanical behavior of natural fibers, especially on their strength and stiffness [13,15]. The chemical treatment of cellulose to obtain various new functions and properties is pervasive. Recently, some eco-friendly approaches have been reported for the deacidifying consolidation of cellulosic structures to make it suitable for long-term preservation [16,17]. Alkaline treatment (or mercerization) is a widely used chemical treatment for natural fibers. It disrupts hydrogen bonding in the network structure and reduces fiber diameter, thereby increasing the aspect ratio [18]. Different kinds of surface treatments of jute fibers have been reported recently, including silane treatment, alkali treatment, and silane + alkali treatment of jute fibers and their corresponding jute fiber-reinforced composites with epoxy resin as a matrix to study their thermal and mechanical properties using vacuum-assisted resin infusion [19]. In another study, micro-silicone and fluorocarbon, two well-known finishing agents of textiles, were used to enhance the surface properties of jute fabric-reinforced composites with a polyester matrix and their flexural, tensile, and interlaminar shear strength were analyzed [20].

Herein, we treated the fibers chemically before making composites and compared the tensile properties of the produced composites with that of untreated jute fiber composites. Since voids greatly influence composite tensile properties, the volume fractions of voids of the treated jute fiber composites were also measured and compared with those of the corresponding raw jute composites.

## 2. Materials and Methods

### 2.1. Materials

Naturally grown straight and long raw jute fibers (*Corchorus olitorius*) were collected from Dhaka, Bangladesh. Epoxy resin (JC-02A) and the corresponding hardener (JC-02B) was sourced from Changshu Jiafa Chemical Co. Ltd, Changshu, China. All the other chemicals for the chemical treatment of jute were purchased from Shanghai Linfeng Chemical Reagent Co. Ltd., Shanghai, China.

### 2.2. Methods

Jute fibers contain unwanted materials known as gum on their surface. These interfere with natural functioning and expected performance. Degumming is a process that is applied to remove impurities. This chemical degumming process involves the sequential steps of alkaline treatment, scouring, and bleaching. Scouring produces the required wettability by removing the gum coating the natural surface. It improves adhesion and locking properties. The experiment was carried out for a combination of acids, oxidizing pretreatment, and scouring. Firstly, the 7-inch samples were cut and slowly combed. Then oven dry weights of the fibers were used for the following processes.

### 2.3. Chemical Treatment

The raw, untreated jute fiber samples are shown in Figure 1a. The fibers were treated in three steps according to a protocol detailed elsewhere [21]. Firstly, samples were treated with 1% H_2_SO_4_ in one bath, followed by treatment with H_2_O_2_, Na_2_SiO_3_, and Na_3_PO_4_ in another bath. Both steps were carried out at 50 °C for 30 min in a bath with a fiber-to-liquor ratio of 1:20. The alkaline treatment is responsible for the disruption of hydrogen bonding in the cellulose. Thus, it increases surface roughness, which is helpful for a more effective interlocking of the fibers and the matrix. This treatment further increases the amount of cellulose exposed, which increases the number of possible reaction sites on the fiber surface. 

In the third step, NaOH along with Na_2_SiO_3_, Na_2_SO_3_, MgSO_4_ (see Table 1) was used to treat the fibers at 80 °C for 90 min at the same liquor ratio used in the previous step and followed by washing three times with distilled water. Samples were then dried in an oven for three hours at 70 °C before composite fabrication, the chemically treated jute fiber samples are shown in Figure 1b.

### 2.4. Composite Preparation

Unidirectional jute fiber composites were prepared by the hand lay-up technique. Composite specimens with different fiber loading (8, 10, 12 wt.%) were prepared with an epoxy resin and hardener ratio of 5:4. The corresponding volume fractions of the fibers were 7%, 8.5%, and 10%, respectively. The specimens were then subjected to curing (at 90 °C for 2 h followed by 110 °C for 1 h and 130 °C for 4 h) and post-curing (at room temperature for 24 h). Composites of both treated (refer, Figure 1d and raw jute (refer, Figure 1c) were prepared, with the latter considered a control.

### 2.5. Characterization

The untreated and treated fabrics were characterized by Attenuated Total Reflectance Fourier Transform Infrared (ATR-FTIR) spectroscopy using Nicolet 6700 spectrometer (Thermofisher, Waltham, MA, USA). A scanning electron microscope (SEM) TM 3000 (Hitachi, Tokyo, Japan) with an acceleration voltage of 15–18 kV and working distance of 10 mm was used to observe the surface morphology of the raw jute fibers, pretreated jute fibers, scoured jute fibers, and the cross-sections of the broken parts of both types of composite after tensile experiments. For surface examination, 1000 times magnification was used and the samples were sputter-coated with gold before observation.

### 2.6. Void Fraction of the Composites

The void fraction (*V_fr_*) of the composites was calculated by using Equation (1)
(1)$$V_{fr}=\frac{\rho_{t}-\rho_{\exp}}{\rho_{t}},$$
where *ρ*_t_ and *ρ*_exp_ are the theoretical and experimental densities of the composite, respectively. The theoretical density of the composites was calculated using Equation (2)
(2)$$\rho_{t}=\frac{1}{\frac{W_{f}}{\rho_{f}}+\frac{W_{m}}{\rho_{m}}},$$
where *ρ* and W are the density and weight fraction respectively. The subscripts *t*, *f*, and *m* correspond to the composites, fiber, and matrix, respectively. The experimental density of the composites was determined by the water immersion method.

### 2.7. Tensile Properties of the Single Fiber

Tensile properties of a single fiber (tensile strength and elongation at break) were tested with a single fiber strength testing machine XQ-1 (the Shanghai New Fiber Instrument Co., Ltd., Shanghai, China) at standard temperature and humidity with a 40 mm gauge length and 5 mm/min crosshead speed. Fifty specimens were tested for each group, and the mean fiber tensile strength is calculated from the data. Before choosing the specimen, the jute was combed lightly to make it smooth for separation. Tensile strength refers to the maximum tensile load a fiber can withstand. Herein, it is expressed in the unit of cN/tex.

### 2.8. Tensile Properties of the Composites

Firstly, the samples were grouped into two groups: treated and untreated (control). The tensile test was performed and the load recorded as per ASTM D 3039 test standards [22]. Herein, we mounted the regular flat strip composites of uniform dimensions (length was 250 mm, width was 25 mm, and thickness was 3 mm), which were loaded monotonically in tension into the grips of a universal mechanical testing machine Instron 1195 with a maximum load capacity of 100 kN (see Figure 1e). The tensile coupons were also gripped by an extensometer to measure the real extension values and avoid variation due to the slight slippage of the samples from the grips. The strain values obtained using the extensometer were used to determine the initial modulus of the samples (see Figure 1f). In this research, five samples from each group were tested, and the one-way analysis of variance with Tukey’s pairwise multiple comparisons was used for statistical data analysis in Minitab. The confidence interval was set at 95%, which means that a P value smaller than 0.05 was considered to be a statistically significant difference. Different letters labeled on the figures indicate significant differences between groups. The error bars shown in figures represent standard deviations.

## 3. Results and Discussion

### 3.1. Physical Properties of the Treated Fiber and the Composites

Not many significant changes were found in the jute fiber surfaces and their physical properties with the elimination of gum (weight loss of around 25% after chemical treatment). This suggests that the fiber was not deteriorated during the chemical treatment. However, analysis of the SEM images shown in Figure 2 suggested that the critical structure of jute became prominent after the removal of impurities through chemical treatment.

### 3.2. FTIR Analysis

FTIR analysis was carried out to estimate the effect of chemical treatment on the jute fibers. The characteristic peak at 3352 cm^−1^ in Figure 3 represents the presence of O–H group stretching in the cellulose of raw jute fiber. The typical peak at 3339 cm^−1^ refers to O–H stretching of the hydrogen bond in the treated jute. The FTIR analysis demonstrates that during the surface modification, this OH peak is not altered much.

However, another characteristic peak at 1731 cm^−1^ of C=O stretch in the ester of the untreated sample disappeared in the treated sample. The disappearance of this peak indicates the removal of waxes and other impurities from the jute fiber surface. The removal of this layer resulted in better interaction adhesion between the matrix and the fiber [23]. The characteristic peak at 2900 cm^−1^ refers to C–H stretching in methyl and methylene groups in the cellulose and hemicellulose of both raw and treated jute fibers. The methyl group (–CH_3_) of lignin might be attributed to the characteristic peak observed at 1456 and 1510 cm^−1^. In addition, the characteristic peaks at 1042 cm^−1^ are assigned to C–O stretching in the glycosidic linkage of the cellulose. Thus, the chemical composition of the samples shows no variation in the chemical bonds, indicating that after chemical treatment, the chemical composition of the jute fibers did not change.

### 3.3. Tensile Properties of Single Fibers

Tensile properties depend on the structure of fibers and have a significant influence on end products. If the strength of jute fiber is high, then the composites will demonstrate high strength. Elongation at break refers to the elongation a fiber exhibits at the maximum breaking load. Table 2 compares the tensile strength and elongation at break of the treated and untreated (raw jute) fibers. Increases of about 4.4% and 6% in tensile strength and elongation at break, respectively, were observed. In this work, the elongation of jute is around 1.8% [24]. Improvement in tensile strength could be attributed to the removal of a more significant flaw leaving a small cross-section area of treated jute fiber. Therefore, it is expected to be stronger than that with a broad cross-section area and a more substantial deficiency (maturity, diameter, and fineness vary from fiber to fiber and area to area). Moreover, it might be attributed to an improved molecular orientation along the loading axis, i.e., fiber longitudinal direction.

### 3.4. The Void Content

The void content decreases with an increase in fiber content, as found previously by Mishra et al. [25]. However, the critical fact is that higher volume fractions of voids were found in composites with untreated jute fibers as compared to treated ones (see Table 3). This finding suggests that compared to raw jute fibers, the treated fibers allowed the epoxy resin to penetrate more quickly into the inter-fiber gaps of the unidirectional fibers, consequently yielding fewer voids.

### 3.5. Tensile Properties of the Composites

All of the jute fiber composites demonstrated a higher tensile strength (>43 MPa) than the pure epoxy one. The tensile strength for both the raw jute composite and the treated jute composite seems to increase with the fiber weight percentages. The maximum load that material may carry before failure is referred to as the ultimate strength of the article. Since the composites contain unidirectional fibers, the variation in tensile strength might be attributed to another vital aspect. Typically, properties of unidirectional composites tested in the fiber direction are mainly influenced by the fiber properties. Therefore, we have tested the tensile properties in the fiber direction in this study. However, the tensile strength of treated jute fiber composites was found to be higher than that of their corresponding raw jute fiber composites, as shown in Figure 4. The lower void fraction and higher volume fractions might be responsible for the higher tensile strength of treated jute composites. The stress-strain curves were plotted using the values obtained by the displacement transducers as shown in Figure 5. Moreover, removal of gum (through alkali treatment) might have resulted in more top surface area of treated jute fibers than the raw jute fibers and consequently created more opportunity for the formation of van der Waals forces between the treated fiber and the epoxy. This can also be evidenced by Figure 6 where the treated jute fiber composite shows lower elongation at break than the raw one at each fiber wt.%. This behavior suggests poor bonding of epoxy resin with the weak spots of the natural (untreated) jute fibers due to the presence of impurities.

With the inclusion of the extensometer, the initial modulus of the composites was calculated from the first 100 readings of the corresponding stress and strain curves for each sample (n = 5). It was found that the chemically treated jute fiber composites, i.e., epoxy-MJ8%, epoxy-MJ10%, and epoxy-MJ12% show a 230%, 113%, and 22% higher initial modulus, respectively, than the corresponding raw jute samples as shown in Figure 7. This might be attributed to better fiber–matrix interaction adhesion. The scoured jute fibers might have a more open structure. A significant improvement in initial modulus was observed for the low fiber content range. Thus, there might be an opportunity to achieve higher composite stiffness with minimal jute fiber content in the composite if the fibers are scoured before being used in polymer composites.

### 3.6. Fractured Surfaces for Epoxy–Jute Fiber Composites

SEM images of the fractured surface of epoxy–jute fiber composites provides direct proof of fiber–matrix interaction adhesion improvement. As shown in Figure 8A, there was no cohesion between the fibers and the matrix, even despite a small amount of epoxy introduction into the interface region. The failure mechanism for this composite was fiber–matrix debonding. This indicates that there was a lack of attraction between the jute fibers and the epoxy, confirming the necessity of enhancing the fiber/epoxy interface. After the treatment, the adhesion interface between the jute fiber and the epoxy is improved. This can be seen by the disappearance of boundary gaps between the epoxy and fiber surface, as indicated by the arrow in Figure 8B. The improvement can be explained by a strong interfacial interaction after the treatment of jute fiber.

## 4. Conclusions

This work discusses the impact of the chemical treatment of jute fibers on the physical and mechanical properties of jute fiber composite. Composites prepared from chemically treated jute fibers were found to be better than the raw jute composites in terms of tensile strength, elongation at break, void fraction, and interfacial adhesion. The findings of this work suggest that the chemical treatment of jute fibers could enable better matrix–fiber adhesion. The results obtained point suggest that the chemical treatment of jute fibers results in an improvement in interfacial bonding to the polymer matrix, which consequently improves the tensile properties of the composites. Further studies should be undertaken for higher fiber volume fractions.

## Figures and Tables

**Figure 1 materials-12-01226-f001:**
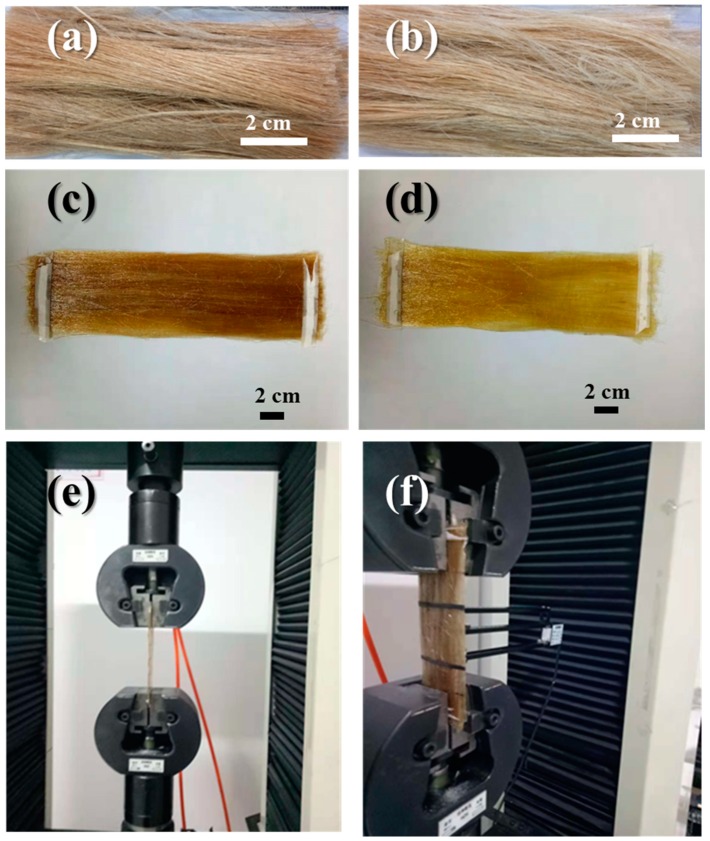
Images of (**a**) untreated (raw jute) and (**b**) chemically treated jute fibers; jute fiber–epoxy composites from unidirectional (**c**) untreated and (**d**) chemically treated fiber and (**e**) front view and (**f**) side view of sample grip arrangement on Instron 1195 tensile testing instrument (Instron, Norwood, MA, USA).

**Figure 2 materials-12-01226-f002:**
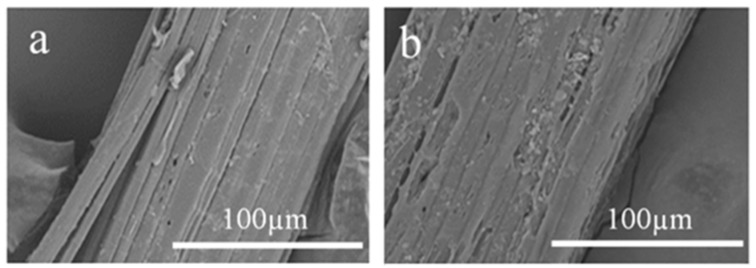
SEM of (**a**) raw and (**b**) treated jute fiber under 1000× magnification.

**Figure 3 materials-12-01226-f003:**
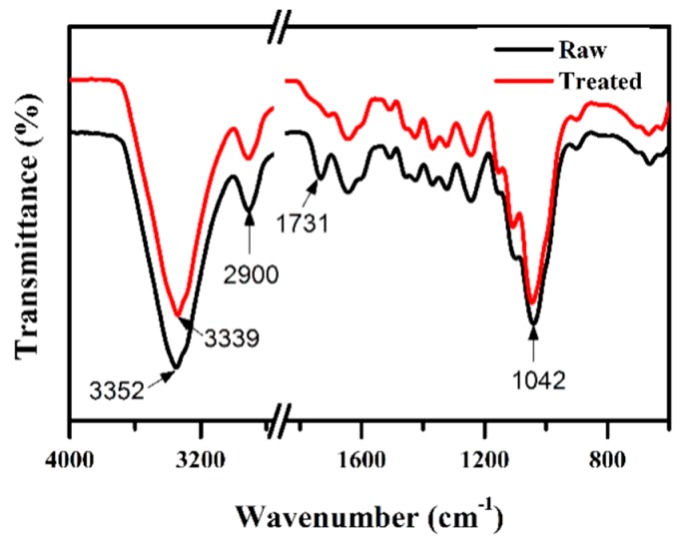
Fourier Transform Infrared (FTIR) spectra of raw and treated jute.

**Figure 4 materials-12-01226-f004:**
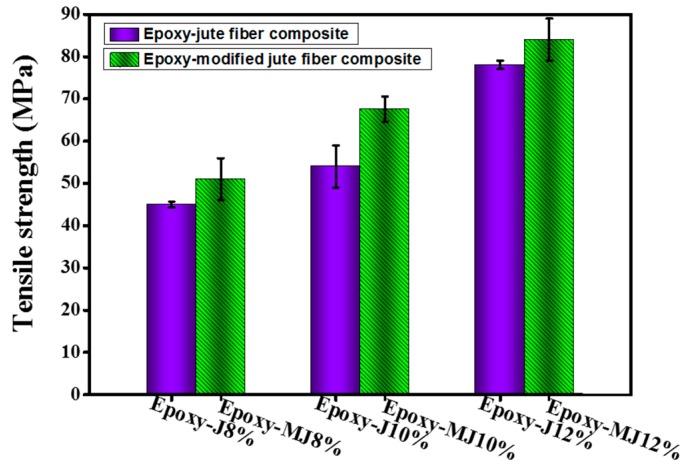
Effect of fiber loading on the tensile strengths of the modified composites.

**Figure 5 materials-12-01226-f005:**
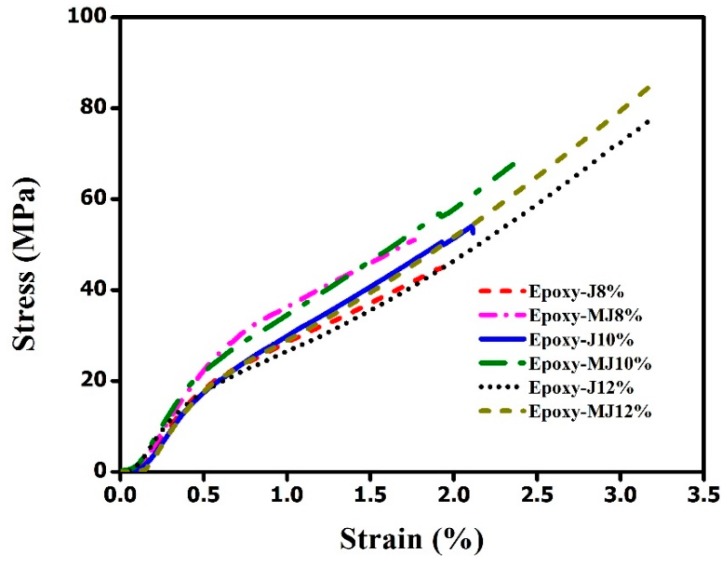
Stress-strain curves of the jute fiber composites with different fiber contents.

**Figure 6 materials-12-01226-f006:**
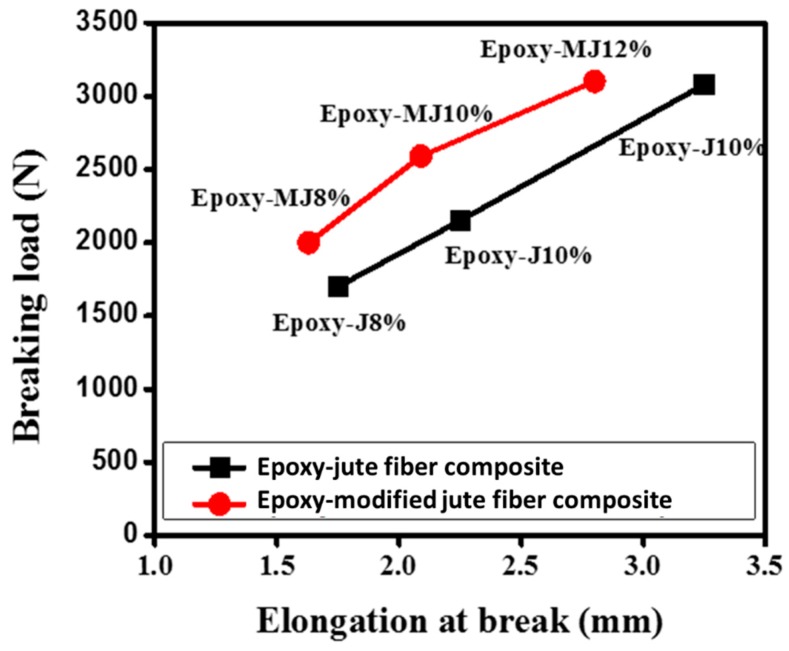
Elongation at break of the jute fiber composites with different fiber contents.

**Figure 7 materials-12-01226-f007:**
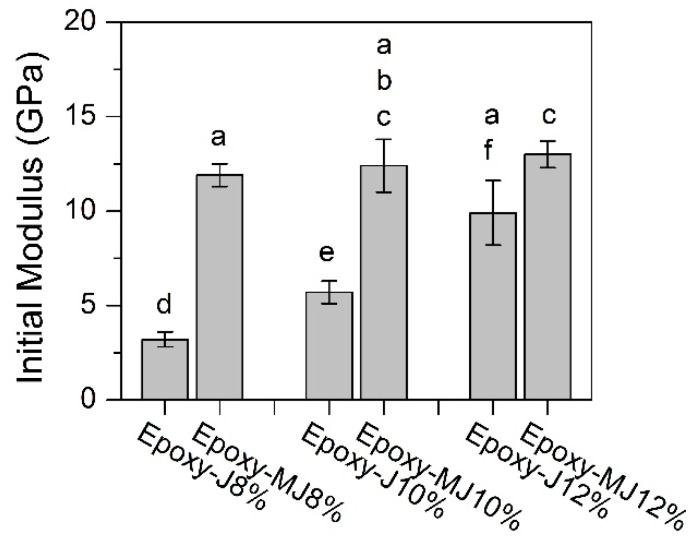
Initial modulus of the composite specimens.

**Figure 8 materials-12-01226-f008:**
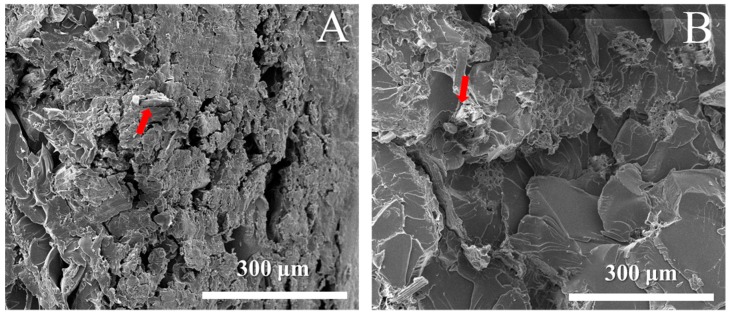
SEM of fractured surfaces (**A**) raw jute (**B**) treated jute composite.

**Table 1 materials-12-01226-t001:** Scouring recipe.

Chemicals	Quantity (g/L)
NaOH	30
Na_2_SiO_3_	3
Na_3_PO_4_	1
Na_2_SO_3_	1.2
MgSO_4_	0.2
JFC	1.5
Emulsifier OP-10	1.5
Peregal O	1
SDBS	1

**Table 2 materials-12-01226-t002:** Tensile strength and elongation at break of single jute fibers.

Jute Fiber	Tensile Strength (cN/tex)	Elongation at Break (%)
Untreated	27.41 ± 0.68	1.51 ± 0.32
Scoured	28.62 ± 1.02	1.63 ± 0.40

**Table 3 materials-12-01226-t003:** Void fraction of different jute fiber composites.

Designation	Composite Composition	Theoretical Density *ρ*_t_ g/cm^3^	Experimental Density *ρ*_ex_ g/cm^3^	Void Fraction (%)
Epoxy-J 8%	Epoxy + 8% raw jute fiber	1.155 ± 0.022	1.101 ± 0.034	4.762 ± 0.001
Epoxy-J 10%	Epoxy + 10% raw jute fiber	1.161 ± 0.023	1.106 ± 0.025	4.737 ± 0.002
Epoxy-J 12%	Epoxy + 12% raw jute fiber	1.165 ± 0.031	1.111 ± 0.031	4.721 ± 0.001
Epoxy-MJ 8%	Epoxy + 8% chemical treated jute fiber	1.157 ± 0.019	1.112 ± 0.021	3.889 ± 0.001
Epoxy-MJ 12%	Epoxy + 10% chemical treated jute fiber	1.164 ± 0.03	1.121 ± 0.023	3.694 ± 0.002
Epoxy-MJ 12%	Epoxy + 12% chemical treated jute fiber	1.17 ± 0.018	1.128 ± 0.024	3.589 ± 0.001

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
