# Peer review of "Effect of Jute Fiber Modification on Mechanical Properties of Jute Fiber Composite"

_materials, 2019, doi:10.3390/ma12081226_

Reviewer 1 Report

The paper describes how modifications of jute fibers change their mechanical properties. It is an interesting and relevant topic. However, major changes would be required before the paper can be published in this journal. 

Introduction:

The first part of the introduction describes the general benefits of composites. However, most of what is written there applies to glass fiber and carbon fiber composites, not necessarily to natural fiber composites. Since this paper covers natural fibers, some statements may be misleading. The authors should address this issue and clearly separate between traditional composites made from manmade materials and natural materials.

More important, this paper covers surface treatment of fibers and subsequent property improvements. Much work has been done in this area and it should be referred to in the introduction or possibly some other place in he paper.

2.3 line 85

… a 1:20 bath. What ratio is that? Please clarify.

2.4 composite preparation, line 102

Please also state the fiber volume fraction. This is what is usually given and allows better comparison to other publications.

2.5 Void fraction

The method given is theoretically OK, but obtaining void fractions of about 4% is questionable.

Please add the measurement error of all individual measurements and the combined error for the void content.

Usually void content is obtained by microscopy. Please add photos and show the shape and location of the voids. It is then also possible to obtain the void content from the photos, which is the usual way of obtaining this value.

3.2 Tensile properties of the fibers

It would be good to show Weibull parameters.

3.3 Void content

Based on the scatter of the density results, there seems to be no difference in void content between treated and untreated fibers.

Only the significant digits behind the decimal point should be listed in the table.

3.4 Tensile properties of laminates

Please show the full stress strain curves of the experiments.

Figure 6 should show stress on the y-axis. Scatter in the stress and strain directions should be indicated.

Why were the first 100 readings used to calculate the modulus? Usually a stress – strain range is chosen. The choice on how the initial modulus is chosen should be justified. It depends typically on the shape of the stress-strain curve.

Discussion

It would be good to have a discussion section putting the work into a larger context.

One aspect that should be discussed: Typically, properties of unidirectional composites tested in the fiber direction are mainly influenced by the fiber properties. 

It should be discussed what laminate strength would be expected based on the fiber volume fraction.

The treatment of the fibers increased the fiber strength by 4%. The laminates have a higher strength increase. A better fiber matrix interface would increase the strength of short fiber composites, but it would not make much difference to long-fiber composites. Also the voids should not influence the strength of a unidirectional long fiber composite too much. Please explain these discrepancies.

It seems that the strength increase is due to other effects than what is described in the paper so far.

Author Response

Comment #1

The paper describes how modifications of jute fibers change their mechanical properties. It is an interesting and relevant topic. However, major changes would be required before the paper can be published in this journal.

Response to reviewer

We are thankful for the reviewer for guiding us by his valuable comments over the manuscript.

Comment #2

Introduction:

The first part of the introduction describes the general benefits of composites. However, most of what is written there applies to glass fiber and carbon fiber composites, not necessarily to natural fiber composites. Since this paper covers natural fibers, some statements may be misleading. The authors should address this issue and clearly separate between traditional composites made from manmade materials and natural materials.

More important, this paper covers surface treatment of fibers and subsequent property improvements. Much work has been done in this area and it should be referred to in the introduction or possibly some other place in the paper.

Response to reviewer

We agree to the reviewer and thus we have rewritten the introduction accordingly.

Comment #3

2.3 line 85 … a 1:20 bath. What ratio is that? Please clarify.

Response to reviewer

We are apologetic for not writing appropriately, we have rewritten it as “in a bath of fiber to liquor ratio of 1:20”.

Comment #4

2.4 composite preparation, line 102

Please also state the fiber volume fraction. This is what is usually given and allows a better comparison to other publications.

Response to reviewer

We have added volume fractions, the corresponding volume fractions of the fibers were 7 %, 8.5 % and 10 %,

Comment #5

2.5 Void fraction

The method given is theoretically OK, but obtaining void fractions of about 4% is questionable.

Response to reviewer

We also agree that 4% of void fraction might be too much, however, this might be due to hand lay-up the process. We can understand that other advanced process can offer less void fraction.

Comment #6

Please add the measurement error of all individual measurements and the combined error for the void content.

Response to reviewer

We have mentioned the combined error for the void content.

Comment #7

Usually void content is obtained by microscopy. Please add photos and show the shape and location of the voids. It is then also possible to obtain the void content from the photos, which is the usual way of obtaining this value.

Response to reviewer

We are apologetic for this, presently we are unable to provide those images. Nevertheless, for the next time, we would try to take those images to present into the manuscript.

Comment #8

3.2 Tensile properties of the fibers

It would be good to show Weibull parameters.

Response to reviewer

We agree with the reviewer to show Weibull parameters, however, they were beyond the scope of this research.

Comment #9

3.3 Void content

Based on the scatter of the density results, there seems to be no difference in void content between treated and untreated fibers.

Response to reviewer

We are apologetic for very less difference between the two. However, some of the void content might be attributed to the hand lay-up method, therefore also the values of void content are a bit higher.

Comment #10

Only the significant digits behind the decimal point should be listed in the table.

Response to reviewer

We agree with your valuable advice, and we have chosen three significant digits behind the decimal point.

Comment #11

3.4 Tensile properties of laminates

Please show the full stress-strain curves of the experiments.

Response to reviewer

We are thankful to the reviewer for this valuable suggestion, we have added full stress-strain curves of the experiments into the revised manuscript.

Comment #12

Figure 6 should show stress on the y-axis. Scatter in the stress and strain directions should be indicated.

Response to reviewer

Since we have presented the stress into Fig 5. Stress-strain curves, therefore, we have kept force on the y-axis into this diagram.

Comment #13

Why were the first 100 readings used to calculate the modulus? Usually a stress-strain range is chosen. The choice on how the initial modulus is chosen should be justified. It depends typically on the shape of the stress-strain curve.

Response to reviewer

Since it is clear from stress-strain curve that the trend of modulus is slightly been changed, therefore, we have relied on the initial modulus of the coupons.

Comment #14

It would be good to have a discussion section putting the work into a larger context.

One aspect that should be discussed: Typically, properties of unidirectional composites tested in the fiber direction are mainly influenced by the fiber properties.

Response to reviewer

We are thankful for the reviewer for his valuable suggestion, we have discussed the influence of fiber direction into the paper.

Comment #15

It should be discussed what laminate strength would be expected based on the fiber volume fraction.

Response to reviewer

We are thankful of the reviewers, we have discussed this into the results and discussion section in the revised manuscript.

Comment #16

The treatment of the fibers increased fiber strength by 4%. The laminates have a higher strength increase. A better fiber-matrix interface would increase the strength of short fiber composites, but it would not make much difference to long-fiber composites. Also, the voids should not influence the strength of a unidirectional long fiber composite too much. Please explain these discrepancies. It seems that the strength increase is due to other effects than what is described in the paper so far.

Response to reviewer

We agree to the reviewer that the improvement by this treatment is low, however, this might be due to mild chemical surface treatment of the jute fibers. Father works, in the future, might be done to do severe surface treatment to see the influence on the mechanical properties.

Reviewer 2 Report

The paper deals with the preparation and characterization of composites based on modified jute fibers. The topic is interesting for readers of Materials. The presentation and discussion of the results could be partly improved. I recommend the publication after the following revisions:

Table 1. The unit for quantity is expressed in gm/L. Is it correct? Please check.

Figure 1. I suggest to report a clear scale bar within the photos.

Figure 2. The scale bar within the photos should be reported.

SEM analysis. Details (working distance, energy of beam,…) should be indicated in the Experimental section.

Figure 4. Please report the label in the y-axis.

Examples of stress vs strain curves should be reported.

I suggest to determine the energy stored up to the breaking by the integration of the stress vs strain curves. Moreover, the elastic modulus could be estimated. These calculations are generally performed on cellulose based composites as reported for reinforced paper (Nanomaterials 2017, 7(8), 199; ACS Appl. Mater. Interfaces, 2018, 10 (32), pp 27355–27364) and hybrid films formed by modified cellulose (Polymers 2019, 11(3), 491). 

Author Response

Comment #1

The paper deals with the preparation and characterization of composites based on modified jute fibers. The topic is interesting for readers of Materials. The presentation and discussion of the results could be partly improved. I recommend the publication after the following revisions:

Response to reviewer

We are thankful for the reviewer to devote his valuable time to review and give his useful comments on our paper.

Comment #2

Table 1. The unit for quantity is expressed in gm/L. Is it correct? Please check.

Response to reviewer

We are sorry for carelessness. We have replaced gm/L with g/L. I hope, it would be fine now.

Comment #3

Figure 1. I suggest to report a clear scale bar within the photos.

Response to reviewer

We have added the clear scale bars within the photos.

Comment #4

Figure 2. The scale bar within the photos should be reported.

Response to reviewer

We have added scale bar within the photos.

Comment #5

SEM analysis. Details (working distance, energy of beam,..) should be indicated in the Experimental section.

Response to reviewer

Following the valuable suggestion, the details such as working distance and acceleration voltage have been indicated in the Experimental section.

Comment #6

Figure 4. Please report the label in the y-axis.

Response to reviewer

We have redrawn the Fig 4, and have reported the label in the y-axis (Now it is as Fig 3)

Comment #7

Examples of stress vs strain curves should be reported.

Response to reviewer

Following the valuable suggestion, we have presented stress vs strain curves in Fig 5.

Comment #8

I suggest to determine the energy stored up to the breaking by the integration of the stress vs strain curves. Moreover, the elastic modulus could be estimated. These calculations are generally performed on cellulose based composites as reported for reinforced paper (Nanomaterials 2017, 7(8), 199; ACS Appl. Mater. Interfaces, 2018, 10 (32), pp 27355–27364) and hybrid films formed by modified cellulose (Polymers 2019, 11(3), 491).

Response to reviewer

We are thankful for you for recommending suitable articles. We found the articles very interesting and worthy and thus have cited them appropriately.

Reviewer 3 Report

As much as I personally prefer to go straight to the testing itself, in this paper it will be better to have a bit more detail in introduction with previous literature.

1.)    thickness of the test specimen were 3 mm – how many layers you have when you made this composite with hand-lay up.

2.)    maybe with some other processes interaction between fibers and matrix will be better. For example, compression moulding, vacuum bagging?

3.)    line 121 – void friction Vfr must be in italic letter.

4.)    line 122 - you need to change the density symbol and write it in italic. See alos line 124.

5.)    maybe to put some figures of production of the composite and tensile testing?

6.)    do you produce some plate and cut out the test specimen in dimension of 45x3 mm? What is the length of test specimens?

7.)    line 129 – what is the max load of the testing machine?

8.)    do you test modulus with mechanical or video extensometer?

9.)    for figure 3 it is missing some more description. Figure 3c is not sharp and there are not any differences between images a, b and c. (it can not be seen)

10.)    line 156 – the authors wrote The characteristic peak at 1731 cm-1 – what characteristic peak?

11.)    what is the difference between sentences in line 164/165 and line 167/168? It is written not understandable.

12.)    These short sentences is a little strange. They should be transformed. Eg. line 165.

13.)    In table 2 – is this average values? I think it will be better to put values of all test specimens and then average and standard deviation

14.)    table 3 put that whole table will be in the same page.

Author Response

Comment #1

As much as I personally prefer to go straight to the testing itself, in this paper, it will be better to have a bit more detail in the introduction with previous literature.

Response to reviewer

We have rewritten the part of the introduction and have gone through detailed review.

Comment #2

The thickness of the test specimen was 3 mm – how many layers you have when you made this composite with hand-lay up.

Response to reviewer

We used totally four layers of fibers when you made this composite with hand layup.

Comment #3

Maybe with some other processes interaction between fibers and matrix will be better. For example, compression molding, vacuum bagging?

Response to reviewer

Yes, each process might have certain advantages over other processes; however, for this study, we have relied only on the hand lay-up method.

Comment #4

Line 121 – void friction Vfr must be in the italic letter.

Response to reviewer

We have changed it to italic.

Comment #5

Line 122 - you need to change the density symbol and write it in italic. See also line 124.

Response to reviewer

We have corrected it in throughout the manuscript.

Comment #6

Maybe to put some figures of production of the composite and tensile testing?

Response to reviewer

We are thankful for your valuable suggestion, following your advice, we have added some figures of production of the composite and tensile testing (see Figure 1)

Comment #7

Do you produce some plate and cut out the test specimen in dimension of 45x3 mm? What is the length of test specimens?

Response to reviewer

We are apologetic for our inaccuracy. We have presented the correct dimensions of the sample for tensile.

Comment #8

Line 129 – what is the max load of the testing machine?

Response to reviewer

We have added the max load of the testing machine, i.e. 100kN.

Comment #9

Do you test modulus with mechanical or video extensometer?

Response to reviewer

Yes, we tested the initial modulus with an electronic extensometer and can be seen in Fig 1C(b).

Comment #10

For figure 3 it is missing some more description. Figure 3c is not sharp and there are not any differences between images a, b and c. (it cannot be seen)

Response to reviewer

We have deleted Figure 3c since it was of no use here. The purpose of SEM was to show that there are not significant deterioration on the surface of jute fiber after chemical treatment.

Comment #11

Line 156 – the authors wrote The characteristic peak at 1731 cm-1 – what characteristic peak?

Response to reviewer

We are apologetic for inattention, we have mentioned that the characteristic peak at 1731 cm-1 is probably from C=O stretch in the ester of the waxes.

Comment #12

What is the difference between sentences in line 164/165 and line 167/168? It is written not understandable.

Response to reviewer

We have rewritten the whole paragraph in the right words.

Comment #13

These short sentences is a little strange. They should be transformed. Eg. Line 165.

Response to reviewer

We have replaced the short sentences, and have made phrases appropriately.

Comment #14

In table 2 – is this average values? I think it will be better to put values of all test specimens and then average and standard deviation

Response to reviewer

Yes, we have shown the arithmetic mean value and have shown the variance in Table 2. We have avoided putting all the values, as it might be too wordy.

Comment #15

Table 3 put that whole table will be on the same page.

Response to reviewer

We have arranged the whole table on the same page.

Reviewer 4 Report

No suggestion to make!

Author Response

Comment #1

No suggestion to make!

Response to reviewer

We are thankful for the reviewer for devoting his/her valuable time and encouraging us to publish this work.

Round  2

Reviewer 2 Report

The paper was improved according to the reviewers' comments. I recommend its publication in the present form.